# Epitope Mapping of Pathogenic Autoantigens on Sjögren’s Syndrome-Susceptible Human Leukocyte Antigens Using In Silico Techniques

**DOI:** 10.3390/jcm11061690

**Published:** 2022-03-18

**Authors:** Shivai Gupta, Danmeng Li, David A. Ostrov, Cuong Q. Nguyen

**Affiliations:** 1Department of Infectious Diseases and Immunology, College of Veterinary Medicine, University of Florida, Gainesville, FL 32611, USA; shivai.gupta@ufl.edu; 2Department of Pathology, Immunology & Laboratory Medicine, College of Medicine, University of Florida, Gainesville, FL 32610, USA; danmeng1986@ufl.edu (D.L.); ostroda@pathology.ufl.edu (D.A.O.); 3Department of Oral Biology, College of Dentistry, University of Florida, Gainesville, FL 32610, USA; 4Center of Orphaned Autoimmune Diseases, University of Florida, Gainesville, FL 32611, USA

**Keywords:** major histocompatibility complex (MHC), human leukocyte antigen (HLA), autoantigens, T cells

## Abstract

Sjögren’s syndrome (SjS) is characterized by lymphocytic infiltration and the dysfunction of the salivary and lacrimal glands. The autoimmune response is driven by the effector T cells and their cytokines. The activation of the effector helper T cells is mediated by autoantigen presentation by human leukocyte antigen (HLA) class II molecules of antigen-presenting cells. Studies using familial aggregation, animal models, and genome-wide association demonstrate a significant genetic correlation between specific risk HLAs and SjS. One of the key HLA alleles is HLA-DRB1*0301; it is one of the most influential associations with primary SjS, having the highest odds ratio and occurrence across different ethnic groups. The specific autoantigens attributed to SjS remain elusive, especially the specific antigenic epitopes presented by HLA-DRB1*0301. This study applied a high throughput in silico mapping technique to identify antigenic epitopes of known SjS autoantigens presented by high-risk HLAs. Furthermore, we identified specific binding HLA-DRB1*0301 epitopes using structural modeling tools such as Immune Epitope Database and Analysis Resource IEDB, AutoDock Vina, and COOT. By deciphering the critical epitopes of autoantigens presented by HLA-DRB1*0301, we gain a better understanding of the origin of the antigens, determine the T cell receptor function, learn the mechanism of disease progression, and develop therapeutic applications.

## 1. Introduction

Sjögren’s syndrome (SjS) is a chronic, systemic autoimmune disease that affects the exocrine glands of the body (salivary and lacrimal glands), which may occur in conjunction with another autoimmune disease [1]. It is estimated that approximately 4 million Americans are affected, making SjS the second most common autoimmune disease after rheumatoid arthritis (RA) [2,3,4]. SjS is a multifactorial disease related to genetic, hormonal, and environmental factors. Based on animal models and candidate gene association studies, susceptibility to developing SjS has been strongly associated with human leukocyte antigen (HLA) class II genes, particularly the HLA-DR and DQ alleles [5]. As with most autoimmune diseases, associations of HLA class II loci with SjS have been described and vary in different ethnic groups [6,7,8,9,10]. In most studies, when an HLA association with primary (p)SjS was demonstrable, a stronger association between HLAs and autoantibody titers could be found to the anti-Ro/SSA and anti-La/SB autoantibody responses. The HLA-DR3 haplotype is associated with SjS and exists within a region with extended linkage disequilibrium not observed in other places in the genome [8]. It is important to note that specific HLA-DR and -DQ alleles have been observed to present autoantigens in SjS (i.e., M3R, α-fodrin, Ro (SSA), and La (SSB)) in different ethnic populations [7,11,12].

The structural determination of disease-relevant peptide–HLA and HLA–peptide–TCR complexes is crucial for the elucidation of the molecular mechanisms responsible for the development of T cell reactivity that promotes autoimmune disease [13]. The HLA–peptide–T cell receptor (TCR) interactions that determine self and non-self-discrimination are guided by a set of rules that malfunction in immunopathology in autoimmunity [14]. There are certain principles that govern HLA restriction and TCR docking geometry with a variety of molecular mechanisms that could affect HLA–peptide–TCR interactions [15]. Autoreactive T cells are directly generated and activated by the mechanisms such as atypical HLA–peptide–TCR binding orientation, low-affinity peptide binding that facilitate thymic escape, TCR-mediated stabilization of weak peptide–HLA interaction, and presentation of peptides in a different binding register [13,16,17,18,19]. The peptide binding register refers to the ∼9-mer window of a peptide that sits directly within the peptide-binding groove at a given time. Alterations in this register, whereby the same peptide binds a peptide-binding groove utilizing a different 9-mer window, have an altered impact on the generation of autoreactive T cells [20]. Further, autoreactive TCRs can bind self-peptide–HLA complexes with a conventional binding topology and a high affinity as seen in type 1 diabetes and multiple sclerosis, thus highlighting the potential role of the peptide binding register in increasing the risk of autoimmune disease [21].

The first HLA class II associations in SjS described were at the DR3 [22,23] and DR2 [23,24] loci in Caucasian populations [25]. Together these two HLA sub-types were shown to account for up to 90% of the MHC association in patients who had SjS, which have been further confirmed in the majority of subsequent studies evaluating northern European cohorts [24]. In 2005, Anaya and colleagues [26] demonstrated that the HLA-DRB1*0301-DQB1*0201 haplotype was associated with pSjS in Latin Americans. The HLA-DR3 allele is one of the predominant alleles in SjS. The purpose of this study was to use a high-throughput in-silico mapping technique to identify antigenic epitopes binding to known risk HLAs of SjS, with significant emphasis on HLA-DR3 allele. Additionally, we sought to investigate the molecular mimicry of the antigenic epitopes by determining the homology to viral and bacterial pathogens that can bind structurally to individual HLAs.

## 2. Materials and Methods

### 2.1. In Silico Binding Affinities of Peptides for HLA-DR3 and Other Risk Alleles

The Immune Epitope Database (IEDB)—La Jolla Institute for Allergy and Immunology (LIAI), La Jolla, CA, USA—hosts a series of machine learning (ML) based tools, each trained on specific datasets of an experimental peptide-MHC binding affinity matrix. These tools encompass the common approaches of ML, namely, linear regression (LR) and utilization of artificial neural networks (ANN). The SMM-align methodology predicted the peptide-MHC binding affinity by fitting a weight matrix that relates peptide sequence to end-point binding affinity value. The datasets used for the IEDB prediction tool SMM-align included complete UniProt protein sequences of human Ro52, Ro60, La, M3R and α-fodrin. Quantitative measurements were selected by choosing the binding assay of identifying the IC50 value, and this was validated by the ΔG measurement values indicating the best possible position the peptide would fit in when being presented to the T cell. The human HLA type II alleles were predicted to bind to 15-mer peptides with a core peptide region of 10 amino acids. Finally, the result sets were analyzed, and the top predicted binders were identified.

### 2.2. PDB Structures of HLA-DR3 and Predicted Peptide Docking

The 1A6A HLA-DR3 MHC II-peptide binding complex was extracted from the Protein Data Bank and was used as a template for the crystal structure in COOT, and geometry regularization in PHENIX modeling (Figure 1, Figure 2, Figure 3, Figure 4 and Figure 5). The peptides were mutated using COOT [27] with rotamers that represent a local energy minimum of torsional angles. The geometry of the resulting complex was regularized in PHENIX. Autodock Vina was used for molecular docking after water, and other atoms were removed, with no presence of peptide [28]. Then, the positions of the peptides with the lowest binding energy (ΔG) were complexed using PHENIX. PyMOL (https://pymol.org/2/) accessed on 15 November 2021 v1.7.2 was used to generate molecular graphic images. The site for docking was prepared by removing all water molecules, and the protonation of HLA-DR3 residues was carried out with the SYBYL-X software. Sets of spheres were used to describe potential binding pockets on the molecular surface of HLA-DR3. The four pockets that were determined for molecular docking were determined using the SPHGEN program [28]. This program generated a grid of points that reflected the shape of the selected site, which were filtered through another program called CLUSTER [29]. CLUSTER grouped the selected spheres to define the points used by the following software called DOCK [29]. DOCK was able to match potential ligand atoms with spheres [28]. The next step used the intermolecular van der Waals and columbic AMBER energy scoring coupled with contact scoring and bump filtering. These additional characteristics were applied to the DOCK program algorithm. Atomic coordinates for all predicted peptides positioned in the selected structural pocket in 1000 different orientations were scored, and based on predicted polar (H bond) and nonpolar (van der Waals) interactions, the best images were obtained. PYMOL was used to generate molecular graphic images.

### 2.3. Homology Determination for Viral and Bacterial Peptides

The software UniProt—Basic Local Alignment Search Tool (https://www.uniprot.org/blast/, accessed on 15 November 2021) was used to find regions of local similarity between sequences with an E-threshold of 10 amino acids in the viral and bacterial databases of UniProt. The top-scoring homology results were verified by the PubMed database (https://blast.ncbi.nlm.nih.gov/Blast.cgi, accessed on 15 November 2021), recorded, and are presented in Tables 14–17. 

## 3. Results

### 3.1. In Silico Antigenic Mapping of High-Risk Autoantigens Presented on HLA-DR3

HLA class II alleles play an important role in the regulation of the immune responses against the Ro and La ribonucleoproteins. The generation of these autoantibodies has been correlated with the alleles DRB1*03:01, DQA1*05:01, and DQB1*02:01 in SjS patients [30,31]. As indicated in Table 1, among all the risk alleles that have been identified in SjS, the DRB1*03:01 allele was found to be a significant risk factor in many ethnicities [32]. According to the European League Against Rheumatism (EULAR) classification criteria for pSjS, the Ro60 antibodies are one of the leading indicators of the onset of disease in patients. DRB1*03:01 haplotype is found to associate with DPB1* 02:01 allele and TNF- α2 alleles in SjS patients [33]. Different HLA alleles can also be protective in nature for varied autoimmune diseases, as presented in Table 2. In DR3 transgenic mice, Ro60 has been shown to induce a strong T and B cell response [34]. Using DRB1*03:01 as the model allele for this study, we sought to map the antigenic epitopes of the five most dominant antigens associated with SjS, which include Ro52, Ro60, La, muscarinic receptor type III (M3R), and α-fodrin as indicated in Table 3. We conducted the mapping using the artificial neural networks of NetMHCIIpan from the Immune Epitope Database and Analysis Resource (IEDB) as a predictive method to identify the 15-amino acid peptides that could potentially be presented by DRB1*03:01. As indicated in Table 4, Table 5, Table 6, Table 7 and Table 8, predicted peptides with the lowest IC50 values, an indicator of half the maximal inhibitory concentration that characterizes the effectiveness of a peptide in substituting a high-affinity molecule for binding to MHC class II represents the binding affinity of that peptide. The lower the IC50 values, the stronger the binding to DRB1*03:01 by the predicted peptides. IEDB primarily uses the threshold of IC50 to select the strong peptide binders respective to the MHC. Peptide binding to MHC class II protein is based on the discrete anchor residues at pockets 1, 4, 6/7, and 9 [35] and these anchor peptide-binding motifs can be used to predict the specific T cell response.

As presented in Table 4 and Figure 1, predicted Ro52 peptides on HLA-DR3 showed the core peptide with an anchor hydrophobic residue leucine at position 1, followed by a negatively charged residue at position 4 with the top-scoring aspartic acid residue and arginine residue at positions 6 and 9. The predicted Ro60 peptides showed a similar trend, in which lysine or phenylalanine (being predominantly a hydrophobic amino acid) was predicted at position 1 followed by a negatively charged residue at position 4, positively charged histidine, lysine, or arginine at position 6, and a positively charged residue at position 9 (Table 5 and Figure 2). The La predicted peptides also indicate the same pattern with isoleucine being present at position 1, followed by a positively charged or uncharged side chain amino acid (e.g., aspartic acid) at position 4 and a polar uncharged side chain at position 6 with a hydrophobic side chain at position 9 (Table 6 and Figure 3). The M3R predicted amino acid 9-mers indicate predominantly hydrophobic side-chained amino acids throughout the entire structure at positions 1, 4, 6, and 9 (Table 7 and Figure 4). Being a 240 KDa protein, α-fodrin showed a similar motif to Ro52, Ro60, and La with a hydrophobic amino acid at position 1 and predominantly charged amino acids at position 4 (predominantly negative) and 6 (predominantly positive), with a hydrophobic amino acid (i.e., lysine) at position 9, as shown in Table 8 and Figure 5.

In summary, in silico antigenic epitope mapping of DRB1*03:01 allele with Ro52, Ro60, La, M3R, and α-fodrin showed that the general trend of all peptides predicted to bind have a backbone structure with position 1 being occupied by a hydrophobic residue, position 4 favors charged amino acids, position 6 favors negatively charged amino acids, and position 9 (especially for Ro52, Ro60) having a positively charged amino acid; α-fodrin was an anomaly preferring a hydrophobic residue at this position. La and M3R mostly indicate hydrophobic residues and amino acids with polar uncharged side chains. This also predicts the nature of the pockets in HLA DRB1*03:01, with positions 1 and 4 being rigid, whereas flexibility in the presentation of amino acids on positions 6 and 9 with either charged or hydrophobic amino acids.

### 3.2. Elucidating the Nature of Predicted Peptides Presented on Other Risk Alleles

As presented in Table 1, in addition to the HLA-DRB1*03:01 allele, there are other pertinent risk HLA alleles that were shown to associate with SjS. To further characterize the antigenic epitopes, we selected five different predominant alleles, specifically HLA-DRB1*01:01, HLA-DRB1*15:01, HLA DRB1*04:05, HLA-DRB4*01:01, and HLA-DRB3*01:01. As indicated in Table 9, Ro52 with the same trend of hydrophobic and charged peptides indicates a strong predictive binding by the NetMHCIIPan for the HLA-DRB1*01:01 and HLA DRB1*04:05 alleles with IC50 values that are lower than 50 nM. Most Ro60 potential binders showed higher IC50 predicted scores for most peptides identifying them to be poor binders (Table 10). La predicted peptides point toward having a slightly different amino acid composition for predicted peptides, with the second anchor position being primarily hydrophobic instead of negatively charged (Table 11). M3R peptides showed a wide disparity in predicted peptide binding for some alleles, suggesting that M3R antigens may be selectively processed and presented based on the presence of alleles such as HLA DRB1*04:05, HLA- DRB1*15:01, and HLA-DRB1*01:01 (Table 12). Lastly, α-fodrin peptide analysis indicates a slightly different sequence of peptides on most risk alleles, as indicated in Table 13. Following a similar pattern to the peptide composition presented, with slight deviations in HLA-DRB3*01:01 and HLA DRB1*04:05, it was observed that HLA-DRB1*01:01 and HLA- DRB1*15:01 had similar peptide presentation patterns to HLA-DRB1*03:01, indicating the higher probability of these alleles presenting the same peptides. In summary, in silico antigenic epitope mapping of HLA-DRB1*01:01, HLA-DRB1*15:01, HLA DRB1*04:05, HLA-DRB4*01:01, and HLA-DRB3*01:01 alleles with Ro52, Ro60, La, M3R, and α-fodrin showed that a similar trend of positions 1 and 4 having hydrophobic and positively charged residues but positions 6 and 9 being fluid to present either a charged or a hydrophobic amino acid for most predicted peptides.

### 3.3. Homology of Predicted Peptides Binding to HLA-DRB1*03:01 to Viral and Bacterial Proteins

Molecular mimicry is one of the main mechanisms by which infections might trigger autoimmune disease [69]. Several viruses and bacteria have been implicated as potential etiological agents in human patients, and specific viruses were determined to cause various clinical signs of SjS in animal models. However, there is still little information about the causative role in disease initiation and progression [70]. As presented, we have identified specific antigenic epitopes of the DRB1*03:01 allele with Ro52, Ro60, La, M3R, and α-fodrin proteins in silico. To determine whether these antigenic epitopes mimic viral and bacterial proteins, we utilized the BLAST tool to identify the amino acid homology between the SjS-associated antigenic epitopes of HLA-DRB1*03:01 and all known viral proteins in the Uniprot databases. As presented in Table 14, Ro52 peptides showed similarities between bat viruses such as *Miniopterus schreibersii* polyomavirus, and other plant-based pathogens. Ro60 peptides showed 100% homology between *Botrytis* (gray mold) viruses which have been stipulated to infect *Botrytis* (a major agricultural hazard) [71]. M3R peptides showed 88.9% homology between a variety of plant-based viral pathogens and affected the growth of agriculture and horticulture-based fungi (pests). La and human/mouse α-fodrin peptides indicate a similarity between *Helenium* virus and *Caudovirales* phages that belong to the family of multiple *Carlaviruses* that infect various ornamental plants [72]. As presented in Table 15 for bacterial proteins, Ro52 predicted core peptides showed 100% homology to *Stigmatella aurantiaca* and *Cystobacter fuscus* that are naturally occurring and a promising source for the discovery of new biologically active natural products [73,74]. Ro60 peptides did not indicate homology to any known bacterial peptides as represented. M3R peptides present a likeness with *Desulfobacterales bacterium* that has a sulfur-based metabolism [75]. La peptides showed 100% similarity between *Pseudomonas* species which is known to cause pneumonia and infections in blood [76], while α-fodrin peptides are homologous to certain aquatic and terrestrial bacteria with an 88.9% similarity. In summary, the results suggest that several environmental factors may be involved in the pathogenesis of SjS, with the main role being played by infectious agents for animals or plants, with molecular homologs acting as triggers that may contribute to disease progression in the existence of a predisposing genetic background. 

### 3.4. Homology of Predicted Peptides Binding to Other Risk HLA Alleles to Viral and Bacterial Proteins

As indicated previously, we have also identified antigenic epitopes of HLA-DRB1*01:01, HLA-DRB1*15:01, HLA-DRB1*04:05, HLA-DRB4*01:01, and HLA-DRB3*01:01 alleles with Ro52, Ro60, La, M3R, and α-fodrin. To further determine if these antigenic epitopes mimic any known viral or bacterial proteins, we compared these peptide sequences using the Uniprot databases. As presented in Table 16, Ro60 predicted peptides of HLA-DRB1*01:01 showed homology with the RNA replication protein of *Botrytis* virus X. *Salmonella* phage SPFM12 showed a similarity to La peptides. In contrast, the M3R peptides indicated a 100% similarity with the *Bacillus* phage. While the α-fodrin peptides for this allele did not indicate a homology with any viral proteins, they were very similar to naturally occurring bacteria that are responsible for fermentation, such as *Candidatus pseudoramibacter* [77] and *Eubacteriaceae bacterium,* which is a pathogen that has been recently found to contribute to colorectal cancer initiation via promoting colitis [78]. HLA- DRB1*15:01 exhibited a homology for either bacteria or viruses for Ro52, Ro6,0, and La. Still, it yielded a 100% homology to the viral protein u (Vpu) protein of the human immunodeficiency virus 1 (HIV-1) to the M3R peptide (IIGNILVIV). Furthermore, HLA- DRB1*15:01 allele is predicted to present peptide EVLDRWRRL, which is very similar to many proteins found in *Streptomyces* and *Saccharopolyspora* species which have been investigated extensively for their bioactive natural pharmacological products [79]. The allele HLA-DRB4*01:01 was shown to present RFLLKNLRP peptide of Ro52. This specific peptide showed a 100% homology between the glycoprotein 120 (gp120) of HIV-1. The RFLLKNLRP peptide of Ro52 also showed homology with many phages and other viruses. Lastly, TYYIKEQKL peptide of Ro60 showed a similarity between the Ro-like RNA binding protein for the *Streptomyces* phage. 

Compared to the bacterial proteins (Table 17), we found that *Helicobacter* sp. showed a 100% homology with the Ro60 peptide IKLLQAQKL. *Helicobacter* sp. has been found to cause chronic gastritis and plays an important role in peptic ulcer disease, gastric carcinoma, and gastric lymphoma. In addition, the homology between Ro60 peptide TYYIKEQKL of HLA-DRB4*01:01 and *Fusobacterium necrophorum*, a rare causative agent of otitis and sinusitis, indicates the linkage of an oral biology homologue [80]. HLA-DRB3*01:01 for both Ro52 and La peptides showed *Virgibacillus massiliensis* and *Oscillospiraceae* bacterium, which have been isolated from the human stool and may form a part of the microbiome, had a 100% homology with CREDLHILF (La) and KRADWKEVI (Ro52) [81,82]. The α-fodrin peptide IKLLQAQKL was indicated to be 100% similar to the glycosyltransferase protein of both *Eubacteriaceae bacterium* and *Candidatus pseudoramibacter* [83], which are microbes that have been observed in the gut [77]. Alterations in the gut and oral microbiota composition have previously been suggested as possible environmental factors in the etiology of pSjS and SLE [84]. In summary, the results suggest that different species in Table 17 belong to *Bacteriodes, Actinomyces*, and *Lactobacillus* that have been found in patients of both pSjS and SLE [85,86,87,88]. In conclusion to the results observed for bacterial homology, it is known that pSjS patients have less diversity in their gut microbiome with less abundant beneficial bacteria and more abundant opportunistic bacteria with pro-inflammatory activity compared with healthy individuals. Out of the primary homologs observed, most of them indicate a 100% homology to the three main bacterial species found in the gut, indicating the gut microbiome contribution in disease progression by molecular mimicry on a genetically predisposed background.

## 4. Discussion

HLA genes are the best documented genetic risk factors for the development of autoimmune diseases and could be directly involved in SjS [89]. This study shows the presence of a similar pattern of amino acids that may be presented by the HLAs based on their structure. The similarity and overlap in the peptides presented on different risk alleles suggest that the same antigenic peptides may be responsible for presenting different autoantigens and thereby initiating the autoimmune cascade. In addition, the results provide insight towards not only the genetic predisposition but also environmental and biological factors that contribute to the onset and progression of the disease. The peptide homology represents similarities in peptides presented to the immune system that shows homology to viral pathogens and bacteria that are both environmental triggers. Bacteria form part of the microbiome of an individual. 

Different amino acids present at specific positions in the biochemical structure may confer protection in the peptides presented. It has been shown in previous studies that, consistent with our results, the requirements of peptides for binding to HLA-DR3 vary among different DR3 binding peptides [90]. Similar to our results, the anchor peptides at different positions 1, 4, 6 and 9 indicate the absence of an anchor or the presence of only a weak anchor residue at either position 4 can be compensated for by the presence of a strong, positively charged anchor residue at position 6 in case of both viral antigens and autoimmune peptides [90,91]. Similar to the predicted peptide trend indicated, Verhagen et al. [92] showed that most insulin and pro-insulin peptides presented in type 1 diabetes also show a similar trend of hydrophobic residues at key anchor positions with a mix of charged residues preferred at other anchor locations. In Graves’ disease, arginine (a positively charged amino acid) has previously been reported to confer a high risk if present at a specific position (in the case of the processed peptide presentation), highlighting the importance of specific residues being present at specific positions for the onset of disease [93,94].

Additionally, we examined certain HLA alleles’ protective role that reduces the probability of specific antigen presentation. HLA-DRB1*01 allele has been proven to be negatively associated with pSjS, a result consistent with the Hungarian population in the study carried out by Kovacs et al. [95]. The protective role of the DRB1*01 allele was confirmed by a meta-analysis in which serological groups DR1 and DR7 were negatively associated with pSjS. However, further research is required in the area [32].

Investigating the cross-presentation of the autoantigen epitopes with bacteria or viruses can provide an important insight into a potential mechanism of disease initiation. The results showed predicted peptides of the five autoantigens exhibiting 100% homology to various reported gut commensal and oral bacteria. Additionally, viral infectious agents that may mimic SjS include hepatitis A, B or C, parvovirus B19, dengue, Epstein Barr virus (EBV), and HIV. Certain viruses express tropism for salivary and lacrimal glandular tissue, especially the herpesvirudae family, which is a large family of DNA viruses that includes cytomegalovirus (CMV), EBV, and human herpesvirus (HHV)-6,7,8. Several lines of epidemiological, serological, and experimental evidence implicate retroviral infections—especially human T-lymphotropic virus type (HTLV)-1, HIVs, human intracisternal A-type retroviral particle (HIAP)-I, and human rhinoviruses (HRV)-5—as triggering factors for the development of SjS. The gut is the most abundant site for bacteria, with nearly 1000 species having microbes that belong to four major phyla: Firmicutes, Bacteroidetes, Actinobacteria, and Proteobacteria. Bacteroidetes, along with Firmicutes, represent more than 90% of the entire plethora of microbes in the gut. Based on our findings, the bifunctional ligase/repressor protein of *Firmicutes bacterium* indicated a 100% homology for a peptide from α-fodrin for the allele HLA- DRB1*15:01. *Eubacteriaceae bacterium, Pseudoramibacter,* and other Firmicutes bacteria’s glycosyltransferases are indicated to have perfect homology to a predicted α-fodrin peptide for the allele HLA-DRB4*01:01. There are indications of multiple *Candidatus* bacterial species, which all belong to the *Firmicutes* phylum for multiple predicted peptides in different alleles, as observed in Table 15 and Table 17. Most indicated bacteria in the data presented had been found to be from three phyla *(Firmicutes*, *Bacteroidetes*, *Actinobacteria*) mentioned above that indicate the probability of bacterial peptides being similar to predicted salivary and lacrimal gland-based proteins that are presented on HLA’s and result in inflammation.

In this study, we were able to predict antigenic epitopes or pathogenic peptides that may be presented in SjS based on a structure-based approach for the HLA cell surface protein. The finding may refine the etiology of the autoimmune process. As simplified in Figure 6, the disease progression is initiated by an environmental trigger like a viral infection on a genetically susceptible individual with a specific HLA allele. Salivary gland epithelial cells experience increased apoptosis and act as sources of pro-inflammatory cytokines such as IFN-γ. Macrophages are attracted to the region and act as the main agents for phagocytosis by participating in tissue destruction. Presentation of viral/bacterial antigens by MHC molecules on antigen-presenting cells leads to priming CD4^+^ T cells. With the help of T cells, B cells can form lymphocytic infiltrates or participate in ectopic germinal center formation where they can undergo class switching, affinity maturation, and differentiation into plasma cells that secrete high levels of antibodies. These antibodies may be cross-reactive against autoantigens such as Ro52, Ro60, La, α-fodrin, and M3R. The autoantibodies can form immune complexes by binding autoantigens and fixing complement or engaging Fc-γ receptors, further facilitating apoptosis. This process results in inflammation and tissue destruction through the recruitment of inflammatory cells and phagocytes to tissues. Apoptotic cells from damaged tissues can be taken up by phagocytes, which present novel autoantigens, supporting further priming and autoreactivity. Therefore, in order to understand the etiology and designing therapies, it is imperative that we understand the genetic factors and the environmental agents working together to create a suitable setting to initiate the autoimmune cascade.

The apparent limitation of the study is that the peptide prediction is strictly based in silico. Since this is an in silico study, the results presented are theoretical and should be subjected to many of the same limitations implicit in the MHC binding affinity prediction tool(s) upon which it is based. Regardless, this is the first study that provides a comprehensive mapping of the antigenic epitopes based on the HLA structure. The advantage of this approach that we describe to map peptides will facilitate in identifying drugs and therapies specific and targeted to disease-susceptible HLA. As listed, many autoimmune diseases are associated with specific HLA alleles and high-resolution crystal structures exist for almost all MHC class II molecules. Strategies for the selection of HLA allele-specific peptides presented and testing their activity in experimental systems can be implemented. Further, this research will aid the ability to identify HLA allele-specific drugs based on the structure that will have applicability for treating autoimmune diseases and other HLA-associated conditions.

## Figures and Tables

**Figure 1 jcm-11-01690-f001:**
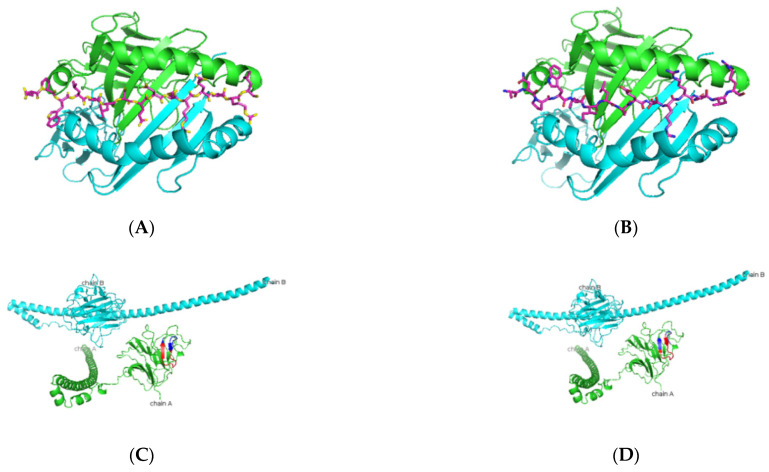
Predicted Ro52 peptide binders docked on HLA-DRB1*0103. (**A**,**B**) Based on the prediction by SMM-align (stabilization matrix alignment) using IEDB, the top two peptides with IC50 values of 28 and 29 were docked onto the crystal structure of HLA-DRB1*0103 PDB structure 1A6A, and the most optimum predicted position of docking is indicated for both peptides-NPWLILSEDRRQVRL and ANPWLILSEDRRQVR with the core sequence of LSEDRRQVR and LILSEDRRQ, respectively. (**C**,**D**) The highlighted region indicates the presence of the peptides in the three-dimensional structure of the Ro52 protein.

**Figure 2 jcm-11-01690-f002:**
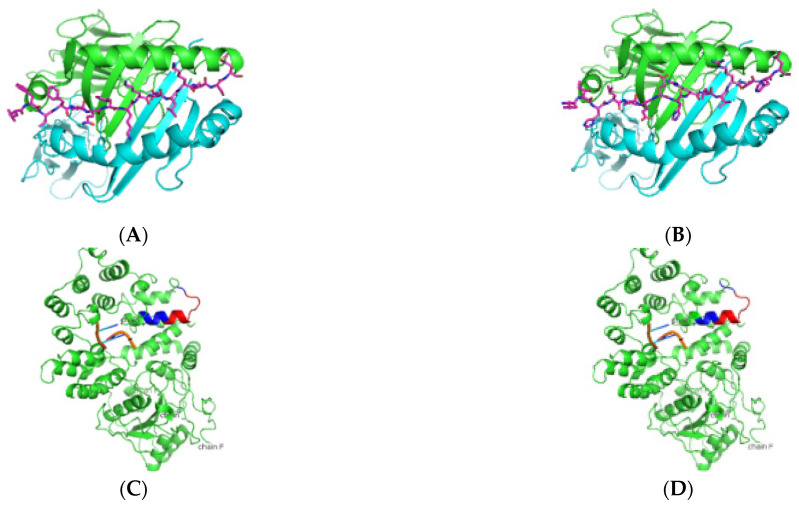
Predicted Ro60 peptide binders docked on HLA-DRB1*0103. (**A**,**B**) SMM-align predicted top two peptides FTFIQFKKDLKESMK and TFIQFKKDLKESMKC with IC50 values of 75 were docked onto the crystal structure of HLA-DRB1*0103 PDB structure 1A6A, and the most optimum predicted position of docking is indicated. Both predicted peptides have the same core sequence FKKDLKESM. (**C**,**D**) The highlighted region indicates the presence of the peptides in the three-dimensional structure of the Ro60 protein.

**Figure 3 jcm-11-01690-f003:**
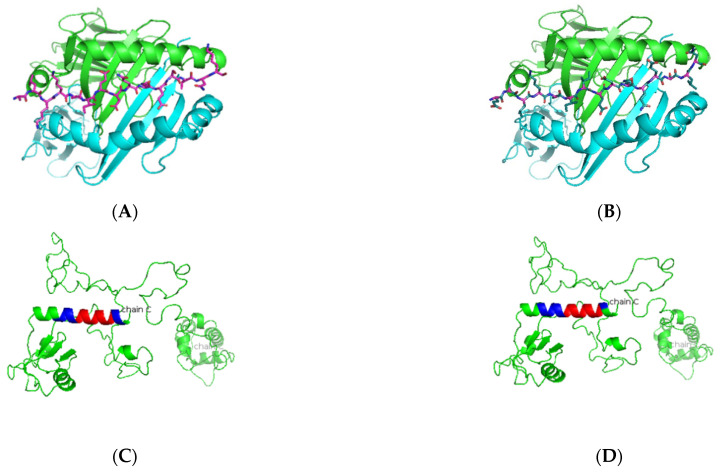
Predicted La peptide binders docked on HLA-DRB1*0103. (**A**,**B**) SMM-align predicted top two peptides ALKKIIEDQQESLNK and EALKKIIEDQQESLN with IC50 values of 49 and 50 were docked onto the crystal structure of HLA-DRB1*0103 PDB structure 1A6A, and the most optimum predicted position of docking is indicated. Both predicted peptides have the same core sequence IIEDQQESL. (**C**,**D**) The highlighted region indicates the presence of the peptide in the three-dimensional structure of the La protein.

**Figure 4 jcm-11-01690-f004:**
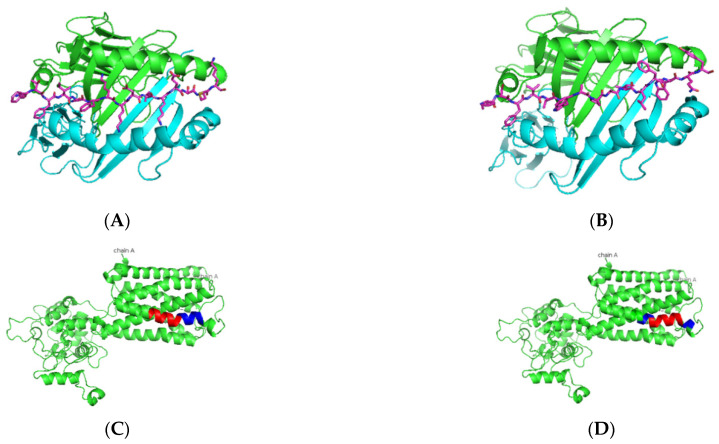
Predicted M3R peptide binders docked on HLA-DRB1*0103. (**A**,**B**) Based on the prediction by SMM-align using IEDB the top two peptides with IC50 values of 120 and 121 were docked onto the crystal structure of HLA-DRB1*0103 PDB structure 1A6A, and the most optimum predicted position of docking is indicated for both peptides AWVISFVLWAPAILF and ISFVLWAPAILFWQY with the core sequence of AWVISFVLW and VLWAPAILF, respectively. (**C**,**D**) The highlighted region indicates the presence of the peptides in the three-dimensional structure of the M3R protein.

**Figure 5 jcm-11-01690-f005:**
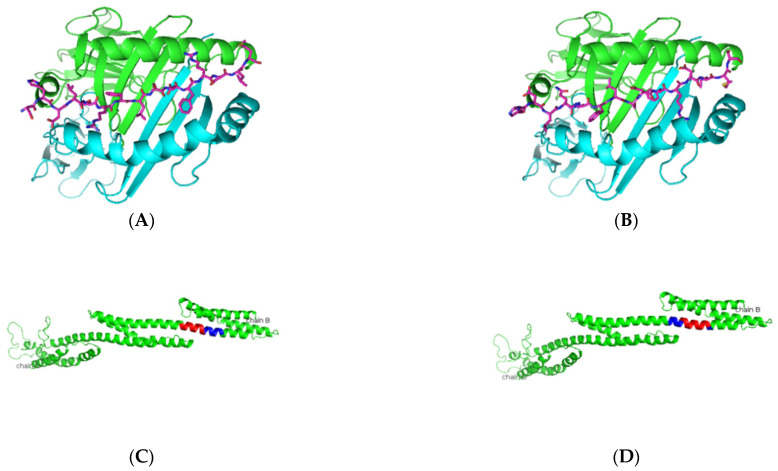
Predicted α-fodrin peptide binders docked on HLA-DRB1*0103. (**A**,**B**) Based on the prediction by SMM-align using IEDB, the top two peptides with IC50 values of 12 were docked onto the crystal structure of HLA-DRB1*0103 PDB structure 1A6A, and the most optimum predicted position of docking is indicated for both peptides SHDLQRFLSDFRDLM and HDLQRFLSDFRDLMS with the core sequence of SHDLQRFLS and FLSDFRDLM, respectively. (**C**,**D**) The highlighted region indicates the presence of the peptides in the three-dimensional structure of the α-fodrin protein.

**Figure 6 jcm-11-01690-f006:**
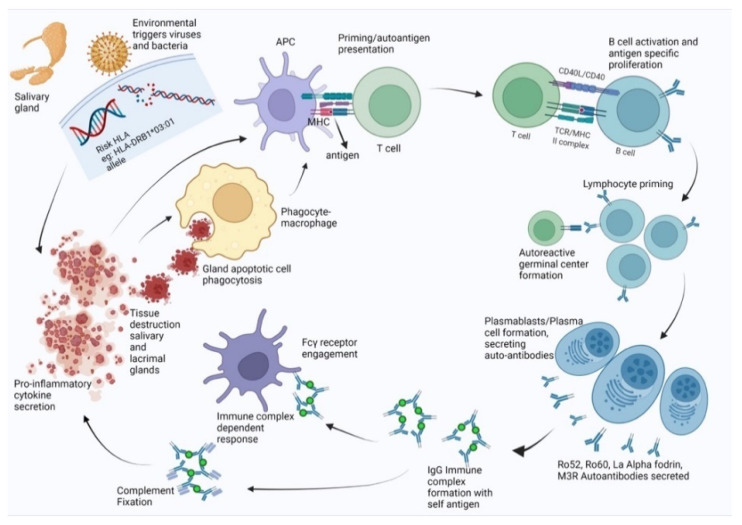
Disease progression for individuals with genetic predisposition (specific HLA) and microbial trigger.

**Table 1 jcm-11-01690-t001:** Identifying high-risk human leukocyte antigen (HLA) alleles in Sjögren’s syndrome (SjS).

Country of Origin/Population	HLA Alleles Connotation	Auto-Antibodies Identified	References
U.S.A./American Caucasian	HLA-B8	ND *	[36]
U.S.A./American Caucasian	HLA-Dw3	ND *	[22]
U.S.A./American Caucasian	HLA-Dw3-HLA-B8	ND *	[12]
U.S.A./American Caucasian	HLA-DRw3-HLA-B8	Antinuclear antibodies Ro60	[37]
U.S.A./American Caucasian	HLA-DRw3-HLA-B8	Ro52	[18]
U.S.A./American Caucasian	HLA-DRw52	SS-A	[38]
Japan/Japanese population	HLA-DRB1*0301	SS-A and SS-B	[39]
	HLA-DRB3*0101		
	HLA-DQA1*0501/DQB1*0201		
Japan/Japanese population	HLA-DRB1*0405	SS-A and SS-B	[39]
	HLA-DRB4*0101		
	HLA-DQA1*0301/DQB1*0401		
Japan/Japanese population	HLA-DRw53	Ro/SS-A and La/SS-B	[40]
Japan/Japanese population	HLA-DRB1*8032/DQA1*0103/DQB1*0601	Ro/SS-A and La/SS-B	[39,41]
	HLA-DRB1*8032		
	HLA-DRB1*0405-DRB4*0101		
	HLA-DQA1*0301		
	HLA-DQB1*0401		
China/Chinese population	HLA-DRB1*0803	SS-A and SS-B	[39]
	HLA-DQA1*0103/DQB1*0601		
Mexico/Mexican population	HLA-DRB1*01:01	Ro/SS-A and La/SS-B	[42]
	HLA-B*35:01		
Colombia/Mestizo Colombian population	HLA-DRB1*0301	Ro/SS-A and La/SS-B	[26,32,43]
	HLA-DQB1*0201		
Israel/Israeli Jewish/Greek	HLA-DQA1*001	SS-A, and SS-B	[44]
	HLA-DQA1*0201/DQB1*0501-Jewish		
	HLA-DQA1*0501-Greek		
Greece/Greek population	HLA-DRB1*0301	Ro/SSA and anti-La/SSB	[45]
Spain/Spanish population	HLA-Cw7	Ro/SSA and anti-La/SSB	[46]
	HLA- DRB1*0301		
	HLA-DR11		
France/French population	HLA-DRB1*1501	ND *	[33]
	HLA- DRB1*0301		
	HLA-DQB1*0201		
	HLA-DQB1*0602		
France/French population	HLA-DRB1*0301	anti-SSA and/or anti-SSB	[31]
	HLA-DQB1*02		
Italy/Italian population	HLA-DRB1*0301	anti-Ro/SSA	[47]
Denmark/Danish population	HLA-Dw2	ND *	[23]
Denmark/Danish population	HLA-DQA1*0501	anti-SSA and/or anti-SSB	[48]
	HLA-DQB1*0201		
	HLA-DQA1*0301		
Finland/Finnish population	HLA-DRB1*0301	anti-SS-A/Ro and anti-SS-B/La	[49]
	HLA-DQA1*0501		
	HLA-DQB1*0201		
Norway/Norwegian Caucasian population	HLA-DRB1*0301	Ro/SSA and La/SSB	[50]
Norway/Norwegian Caucasian population	HLA-DRB1*0301	anti-La/SSB strong positive association with DQA1*0501 anti-Ro/SSA and anti-La/SSB autoantibody response was positively associated with DRB1*03, DQB1*02 and DRB1*03/DRB1*15-DQB1*02/DQB1*0602	[19]
	HLA-DQB1*02		
	HLA-DQA1*0501		
United Kingdom/British Caucasian population	HLA-DRB1*0301	Ro/SSA and La/SSB	[51]
	HLA-DRw52		
Australia/Australian population	HLA-DRB1*0301	Ro/SSA and La/SSB	[16]
	HLA-DQA1*0501		
	HLA-DQB1*02		
Tunisian population	HLA-DQB1 CAR1/CAR2	ND *	[52]
European and African American population	HLA-DQB1*0201	SSA	[53]
	HLA-DQA1*0101		

ND *—not determined.

**Table 2 jcm-11-01690-t002:** Identifying protective HLA alleles in different autoimmune diseases.

Disease	Protective HLA Class II Allele	References
Graves’ disease	HLA-DRB1*07	[54]
	HLA-DQB1*02	
	HLA-DQA1*02	
Hashimoto’s thyroiditis	HLA-DRB1*07	[55]
	HLA-DQB1*02	
	HLA-DQA1*02	
Rheumatoid arthritis	HLA-DRB1*0103	[56]
	HLA-DRB1*07	
	HLA-DRB1*1201	
	HLA-DRB1*1301	
	HLA-DRB1*1501	
Multiple sclerosis	DRB1*14-DQB1*06-DQA1*0102	[9]
Type 1 diabetes	DRB1*14-DQB1*06-DQA1*0102	[57]
	DRB1*15-DQB1*06-DQA1*01	
Systemic lupus erythematosus	DR4	[58]
	DR5	
	DR11	
	DR14	

**Table 3 jcm-11-01690-t003:** Peptides for SjS that have been tested in vivo.

Peptide	Amino Acids	Amino Acid Sequence	In Vivo Confirmation	References	HLA-DR3	IC50
M3R	205–237	LFWQYFVGKRTVPPGECFIQFLSEPTITFGTAI	NOD/LtJ mice	[59]	GECFIQFLSEPTITF	473
	208–227	QYFVGKRTVPPGECFIQFLS	Immunization of young female NOD/LtJ mice on autoimmune response	[60]	QYFVGKRTVPPGECF	8607
		Part of second extracellular loop				
	213–228	KRTVPPGECFIQFLSE	BALB/c	[61]	KRTVPPGECFIQFLS	50,000
	514–527	NTFCDSCIPKTFWN	BALB/c	[61]	NTFCDSCIPKTFWNL	6549
		MTLHSNSTTSPLFPNISSSWVHSPSEAGLP, N1	C57BL/6j (B6) mice (M3R+/+) M3R−/− mice Rag1−/− mice	[62]	PNISSSWVHSPSEAG	4760
		VHSPSEAGLPLGTVSQLDSYNISGTSGNFS, N2			LPLGTVSQLDSYNIS	6028
		NISQTSGNFSSNDTSSDPLGGHTIWQV, N3			TSGNFSSNDTSSDPL	6471
		FTTYIIMNRWALGNLACDLW, Extracellular loop 1			FTTYIIMNRWALGNL	955
		QYFVGKRTVPPGECFIQFLSEP, Extracellular loop 2			QYFVGKRTVPPGECF	8607
		VLVNTFCDSCIPKTYWNLGY, Extracellular loop 3			VLVNTFCDSCIPKTY	5219
	H 441–465	PAGGTDCSLPMIWAQKTNTPADVFI	SJL/L (H-2s) A/J(H-2a)	[63]	TDCSLPMIWAQKTNT	2068
	H 316–335	KARIHPFHILIALETYKTGH	SJL/L (H-2s) BALB/c (H-2d) A/J(H-2a)	[63]	IHPFHILIALETYKT	1485
	H 306–325	EKLCNEKLLKKARIHPFHIL	SJL/L (H-2s)	[63]	EKLLKKARIHPFHIL	1721
	H 26–45	QVTDMNRLHRFLCFGSEGGT	SJL/L (H-2s)	[63]	QVTDMNRLHRFLCFG	2266
	H 401–425	MVVTREKDSYVVAFSDEMVPCPVT	SJL/L (H-2s) A/J(H-2a)	[63]	REKDSYVVAFSDEMV	2879
	H 481–505	IALREYRKKMDIPAKLIVCGMSTNG	SJL/L (H-2s)	[63]	REYRKKMDIPAKLIV	622
	H 201–225	YITKGWKEVHELYKEKALSVETEKL	BALB/c (H-2d)	[63]	VHELYKEKALSVETE	2191
	H 241–265	ELEVIHLIEEHRLLTNHLKS	BALB/c (H-2d) A/J(H-2a)	[63]	VIHLIEEHRLLTNHL	130
Ro52	Full peptide	Full protein	New Zealand Mixed Mice (NZMZ) 2758	[64]	NPWLILSEDRRQVRL	28
Ro60	480–494	AIALREYRKKMDIPA	Animals were immunized with peptide Ro480–494	[65,66]	AIALREYRKKMDIPA	1876
	274–290	QEMPLTALLRNLGKMT	Animals were immunized with peptide Ro274–290	[65,66]	EMPLTALLRNLGKMT	1598
	274–290	Human QEMPLTALLRNLGKMT	Amino acid sequences of the human 60-kd Ro peptides used for immunization of BALB/c mice	[67]	EMPLTALLRNLGKMT	1598
		Mouse QEMPLTALLRNLGKMT				
	413–428	Human VAFSDEMVPCPVTTDM				
		Mouse VAFACDMVPFPVTTDM				
		Rabbit VAFSDEMVPCPLTTDM				
	480–495	Human AIALREYRKKMDIPA			VAFSDEMVPCPVTTD	20,917
		Mouse AVALREYRKKMDIPA			AIALREYRKKMDIPA	1876
La	1–107	GYVDISLLVSFNKMKKLTTDGKLIARALKSSSVVELDLEGTRIRRKKPLGERPKDEEERTVYVELLPKNVTH		[68]	MKKLTTDGKLIARAL	136
	243–345	KAKKRAQKDGVGQAASEVSKESRDLEFCSTEEEKETDRKGDSLSKVKRKHKKKHKERHKMGEEVIPLRVLSKTEWMDLKKEYLALQKASMASLKKTISQ			SKTEWMDLKKEYLAL	922
	111–242	EQAAKAIEFLNNPPEEAPRKPGIFPKTVKNKPIPSLRVAEEKKKKKKKKGRIKKEESVQAKESAVDSSSSGVCKATKRPRTASEGSEAETPEAPKQPAKKKKKRDKVEASSLPEARAGKRERCSAEDEDCL			SSSGVCKATKRPRTA	561

**Table 4 jcm-11-01690-t004:** HLA-DR3 allele with predicted peptides of human Ro52.

Allele	Start	End	Length	Core Sequence	Peptide Sequence	IC50	Percentile Rank	Adjusted Rank
HLA-DRB1*03:01	297	311	15	LSEDRRQVR	NPWLILSEDRRQVRL	28.00	0.10	0.10
HLA-DRB1*03:01	296	310	15	LILSEDRRQ	ANPWLILSEDRRQVR	29.00	0.11	0.11
HLA-DRB1*03:01	298	312	15	LSEDRRQVR	PWLILSEDRRQVRLG	29.00	0.11	0.11
HLA-DRB1*03:01	299	313	15	LSEDRRQVR	WLILSEDRRQVRLGD	29.00	0.11	0.11
HLA-DRB1*03:01	300	314	15	LSEDRRQVR	LILSEDRRQVRLGDT	29.00	0.11	0.11
HLA-DRB1*03:01	301	315	15	LSEDRRQVR	ILSEDRRQVRLGDTQ	90.00	0.91	0.91
HLA-DRB1*03:01	302	316	15	LSEDRRQVR	LSEDRRQVRLGDTQQ	93.00	0.95	0.95
HLA-DRB1*03:01	197	211	15	LEKDEREQL	LQELEKDEREQLRIL	129.00	1.60	1.60
HLA-DRB1*03:01	196	210	15	LEKDEREQL	QLQELEKDEREQLRI	137.00	1.60	1.60
HLA-DRB1*03:01	198	212	15	LEKDEREQL	QELEKDEREQLRILG	137.00	1.60	1.60

**Table 5 jcm-11-01690-t005:** HLA-DR3 allele with predicted peptides of human Ro60.

Allele	Start	End	Length	Core Sequence	Peptide Sequence	IC50	Percentile Rank	Adjusted Rank
HLA-DRB1*03:01	126	140	15	FKKDLKESM	FTFIQFKKDLKESMK	75.00	0.70	0.70
HLA-DRB1*03:01	127	141	15	FKKDLKESM	TFIQFKKDLKESMKC	75.00	0.70	0.70
HLA-DRB1*03:01	125	139	15	LFTFIQFKK	LFTFIQFKKDLKESM	76.00	0.74	0.74
HLA-DRB1*03:01	244	258	15	LIEEHRLVR	VIHLIEEHRLVREHL	78.00	0.76	0.76
HLA-DRB1*03:01	128	142	15	FKKDLKESM	FIQFKKDLKESMKCG	79.00	0.77	0.77
HLA-DRB1*03:01	245	259	15	LIEEHRLVR	IHLIEEHRLVREHLL	80.00	0.77	0.77
HLA-DRB1*03:01	129	143	15	FKKDLKESM	IQFKKDLKESMKCGM	81.00	0.79	0.79
HLA-DRB1*03:01	242	256	15	LIEEHRLVR	LEVIHLIEEHRLVRE	82.00	0.82	0.82
HLA-DRB1*03:01	243	257	15	LIEEHRLVR	EVIHLIEEHRLVREH	82.00	0.82	0.82
HLA-DRB1*03:01	241	255	15	ELEVIHLIE	ELEVIHLIEEHRLVR	83.00	0.83	0.83

**Table 6 jcm-11-01690-t006:** HLA-DR3 allele with predicted peptides of human La.

Allele	Start	End	Length	Core Sequence	Peptide Sequence	IC50	Percentile Rank	Adjusted Rank
HLA-DRB1*03:01	328	15	15	IIEDQQESL	ALKKIIEDQQESLNK	49.00	0.36	0.36
HLA-DRB1*03:01	327	15	15	IIEDQQESL	EALKKIIEDQQESLN	50.00	0.37	0.37
HLA-DRB1*03:01	326	15	15	KEALKKIIE	KEALKKIIEDQQESL	51.00	0.38	0.38
HLA-DRB1*03:01	329	15	15	IIEDQQESL	LKKIIEDQQESLNKW	51.00	0.38	0.38
HLA-DRB1*03:01	330	15	15	IIEDQQESL	KKIIEDQQESLNKWK	54.00	0.40	0.40
HLA-DRB1*03:01	91	15	15	ISEDKTKIR	AELMEISEDKTKIRR	131.00	1.60	1.60

**Table 7 jcm-11-01690-t007:** HLA-DR3 allele with predicted peptides of human M3R.

Allele	Start	End	Length	Core Sequence	Peptide Sequence	IC50	Percentile Rank	Adjusted Rank
HLA-DRB1*03:01	192	206	15	AWVISFVLW	AWVISFVLWAPAILF	120.00	1.40	1.40
HLA-DRB1*03:01	195	209	15	VLWAPAILF	ISFVLWAPAILFWQY	121.00	1.40	1.40
HLA-DRB1*03:01	193	207	15	VLWAPAILF	WVISFVLWAPAILFW	123.00	1.50	1.50
HLA-DRB1*03:01	194	208	15	VLWAPAILF	VISFVLWAPAILFWQ	123.00	1.50	1.50
HLA-DRB1*03:01	196	210	15	VLWAPAILF	SFVLWAPAILFWQYF	125.00	1.50	1.50
HLA-DRB1*03:01	375	389	15	ILNSTKLPS	STILNSTKLPSSDNL	169.00	2.30	2.30
HLA-DRB1*03:01	374	388	15	ILNSTKLPS	HSTILNSTKLPSSDN	170.00	2.30	2.30
HLA-DRB1*03:01	371	385	15	LPGHSTILN	LPGHSTILNSTKLPS	171.00	2.30	2.30
HLA-DRB1*03:01	372	386	15	ILNSTKLPS	PGHSTILNSTKLPSS	171.00	2.30	2.30
HLA-DRB1*03:01	373	387	15	ILNSTKLPS	GHSTILNSTKLPSSD	171.00	2.30	2.30
HLA-DRB1*03:01	548	562	15	FRTTFKMLL	NKTFRTTFKMLLLCQ	198.00	2.70	2.70
HLA-DRB1*03:01	546	560	15	FRTTFKMLL	LCNKTFRTTFKMLLL	199.00	2.70	2.70
HLA-DRB1*03:01	549	563	15	FRTTFKMLL	KTFRTTFKMLLLCQC	199.00	2.70	2.70
HLA-DRB1*03:01	547	561	15	FRTTFKMLL	CNKTFRTTFKMLLLC	200.00	2.70	2.70

**Table 8 jcm-11-01690-t008:** HLA-DR3 allele with predicted peptides of human α-fodrin.

Allele	Start	End	Length	Core Sequence	Peptide Sequence	IC50	Percentile Rank	Adjusted Rank
HLA-DRB1*03:01	1318	1332	15	SHDLQRFLS	SHDLQRFLSDFRDLM	12.00	0.01	0.01
HLA-DRB1*03:01	1319	1333	15	FLSDFRDLM	HDLQRFLSDFRDLMS	12.00	0.01	0.01
HLA-DRB1*03:01	1320	1334	15	FLSDFRDLM	DLQRFLSDFRDLMSW	12.00	0.01	0.01
HLA-DRB1*03:01	1322	1336	15	FLSDFRDLM	QRFLSDFRDLMSWIN	12.00	0.01	0.01
HLA-DRB1*03:01	363	377	15	FLADFRDLT	LQRFLADFRDLTSWV	26.00	0.06	0.06
HLA-DRB1*03:01	360	374	15	SYRLQRFLA	SYRLQRFLADFRDLT	27.00	0.07	0.07
HLA-DRB1*03:01	361	375	15	FLADFRDLT	YRLQRFLADFRDLTS	27.00	0.07	0.07
HLA-DRB1*03:01	362	376	15	FLADFRDLT	RLQRFLADFRDLTSW	27.00	0.07	0.07
HLA-DRB1*03:01	364	378	15	FLADFRDLT	QRFLADFRDLTSWVT	28.00	0.10	0.10
HLA-DRB1*03:01	1323	1337	15	FLSDFRDLM	RFLSDFRDLMSWING	36.00	0.16	0.16
HLA-DRB1*03:01	1324	1338	15	FLSDFRDLM	FLSDFRDLMSWINGI	37.00	0.17	0.17
HLA-DRB1*03:01	365	379	15	FLADFRDLT	RFLADFRDLTSWVTE	83.00	0.83	0.83
HLA-DRB1*03:01	366	380	15	FLADFRDLT	FLADFRDLTSWVTEM	85.00	0.83	0.83

**Table 9 jcm-11-01690-t009:** Predicted peptides on risk alleles for human Ro52.

Allele	Core Sequence	Peptide Sequence	IC50
HLA-DRB1*01:01	LKNLRPNRQ	RFLLKNLRPNRQLAN	44.00
	RFLLKNLRP	CRQRFLLKNLRPNRQ	52.00
HLA-DRB1*15:01	TGPLRPFFS	CAFTGPLRPFFSPGF	122.00
	LRPFFSPGF	AFTGPLRPFFSPGFN	123.00
HLA -DRB1*04:05	EAGMVSFYN	LDYEAGMVSFYNITD	39.00
	MVSFYNITD	DYEAGMVSFYNITDH	39.00
HLA-DRB4*01:01	LKNLRPNRQ	RFLLKNLRPNRQLAN	102.00
	RFLLKNLRP	CRQRFLLKNLRPNRQ	110.00
HLA-DRB3*01:01	KRADWKEVI	IAIKRADWKEVIIVL	229.00
	EVEIAIKRA	EVEIAIKRADWKEVI	247.00

**Table 10 jcm-11-01690-t010:** Predicted peptides on risk alleles for human Ro60.

Allele	Core Sequence	Peptide Sequence	IC50
HLA-DRB1*01:01	LFTFIQFKK	LFTFIQFKKDLKESM	286.00
	FKKDLKESM	FTFIQFKKDLKESMK	289.00
HLA-DRB1*15:01	IQEIKSFSQ	CEVIQEIKSFSQEGR	238.00
	VIQEIKSFS	GRGCEVIQEIKSFSQ	251.00
HLA-DRB1*04:05	LRLSHLKPS	HKDLLRLSHLKPSSE	75.00
	LSHLKPSSE	DLLRLSHLKPSSEGK	75.00
HLA-DRB4*01:01	TYYIKEQKL	EGGTYYIKEQKLGLE	228.00
	KDLLRLSHL	SHKDLLRLSHLKPSS	237.00
HLA-DRB3*01:01	LFTFIQFKK	LFTFIQFKKDLKESM	286.00
	FKKDLKESM	FTFIQFKKDLKESMK	289.00

**Table 11 jcm-11-01690-t011:** Predicted peptides on risk alleles for human La.

Allele	Core Sequence	Peptide Sequence	IC50
HLA-DRB1*01:01	FNVIVEALS	TDFNVIVEALSKSKA	52.00
	DFNVIVEAL	NRLTTDFNVIVEALS	60.00
HLA- DRB1*15:01	LHILFSNHG	REDLHILFSNHGEIK	34.00
	DLHILFSNH	QTCREDLHILFSNHG	37.00
HLA-DRB1*04:05	FNVIVEALS	LTTDFNVIVEALSKS	66.00
	NRLTTDFNV	NRLTTDFNVIVEALS	67.00
HLA-DRB4*01:01	EIMIKFNRL	VPLEIMIKFNRLNRL	75.00
	IKFNRLNRL	PLEIMIKFNRLNRLT	77.00
HLA-DRB3*01:01	DLDDQTCRE	DLDDQTCREDLHILF	142.00
	CREDLHILF	LDDQTCREDLHILFS	143.00

**Table 12 jcm-11-01690-t012:** Predicted peptides on risk alleles for human M3R.

Allele	Core Sequence	Peptide Sequence	IC50
HLA-DRB1*01:01	IAFLTGILA	VVFIAFLTGILALVT	9.00
	LTGILALVT	FIAFLTGILALVTII	12.00
HLA- DRB1*15:01	IIGNILVIV	VTIIGNILVIVSFKV	14.00
	ILVIVSFKV	IIGNILVIVSFKVNK	14.00
HLA-DRB1*04:05	VPPGECFIQ	VPPGECFIQFLSEPT	7.00
	FIQFLSEPT	PPGECFIQFLSEPTI	7.00
HLA-DRB4*01:01	LVTIIGNIL	GILALVTIIGNILVI	88.00
	IGVISMNLF	ADLIIGVISMNLFTT	98.00
HLA-DRB3*01:01	GECFIQFLS	GECFIQFLSEPTITF	124.00
	FLSEPTITF	ECFIQFLSEPTITFG	127.00

**Table 13 jcm-11-01690-t013:** Predicted peptides on risk alleles for human α-fodrin.

Allele	Core Sequence	Peptide Sequence	IC50
HLA-DRB1*01:01	FQKIKSMAA	NGRFQKIKSMAASRR	3.00
	IKLLQAQKL	MREKGIKLLQAQKLV	5.00
HLA- DRB1*15:01	WRRLKAQMI	LDRWRRLKAQMIEKR	68.00
	EVLDRWRRL	NEVLDRWRRLKAQMI	71.00
HLA-DRB1*04:05	FRSSLSSAQ	HDAFRSSLSSAQADF	38.00
	HDAFRSSLS	REAHDAFRSSLSSAQ	39.00
HLA-DRB4*01:01	KMREKGIKL	KMREKGIKLLQAQKL	5.00
	IKLLQAQKL	MREKGIKLLQAQKLV	5.00
HLA-DRB3*01:01	IQETRTYLL	IQETRTYLLDGSCMV	25.00
	YLLDGSCMV	QETRTYLLDGSCMVE	25.00

**Table 14 jcm-11-01690-t014:** Homology of predicted peptides binding to HLA-DRB1*03:01 to viral proteins.

Protein	Predicted Peptide	Virus	Protein	Homology with Sequence (Percentage)
Human Ro52	LEKDEREQL	*Miniopterus schreibersii* polyomavirus 1	Large T antigen	88.9%
		*Micromonas pusilla* virus PL1	Uncharacterized	77.8%
		*Miniopterus schreibersii* polyomavirus 1	Small T antigen	88.9%
Mouse Ro52	MEMDLTMQR		*Wiseana iridescent* virus (WIV) (Insect iridescent virus type 9)	70%
Mouse Ro52	KELAEKMEM	*Mimivirus* LCMiAC02	Uncharacterized	77.8%
Mouse Ro60	LFTFIQFKK	*Botrytis* virus X (isolate *Botrytis cinerea*/New Zealand/Howitt/2006) (BOTV-X)	RNA replication	100%
Human Ro60	LFTFIQFKK	*Botrytis* virus X (isolate *Botrytis cinerea*/New Zealand/Howitt/2006) (BOTV-X)B19:B22	RNA replication protein	100%
Human M3R	AWVISFVLW	*Pseudomonas* phage PaMx74	Putative membrane protein	75%
Human M3R	LPGHSTILN	Pepper mild mottle virus (strain Spain) (PMMV-S)	Replicase large subunit	88.9%
		*Odontoglossum* ringspot virus (isolate Korean Cy) (ORSV-Cy)	Replicase large	88.9%
		Tobacco mild green mosaic virus (TMGMV) (TMV strain U2)	Replicase large subunit	88.9%
		Turnip vein-clearing virus (TVCV)	Replicase large subunit	88.9%
		Youcai mosaic virus (YoMV)	Replicase large subunit	88.9%
		Hoya necrotic spot virus	Methyltransferase/RNA helicase	88.9%
		Odontoglossum ringspot virus	Methyltransferase/RNA helicase	88.9%
		*Virgaviridae* sp.	Replication-associated protein	88.9%
		Tobacco mild green mosaic virus (TMGMV) (TMV strain U2)	Replicase large subunit	88.9%
		Brugmansia mild mottle virus	Methyltransferase/RNA helicase	88.9%
		Streptocarpus flower break virus	Methyltransferase/RNA helicase	88.9%
		Ribgrass mosaic virus (RMV)	Methyltransferase/RNA helicase	88.9%
		Wasabi mottle virus	Methyltransferase/RNA helicase	88.9%
		Piper chlorosis virus	Replicase large subunit	88.9%
Human La	KEALKKIIE	*Helenium* virus S (HelVS)	Helicase	88.9%
		*Arthrobacter* phage Boersma	DNA polymerase I	100%
Human/Mouse α-fodrin	SYRLQRFLA	Uncultured Caudovirales phage	Uncharacterized protein	88.9%

**Table 15 jcm-11-01690-t015:** Homology of predicted peptides binding to HLA-DRB1*03:01 to bacterial proteins.

Human Ro52	LSEDRRQVR	*Stigmatella aurantiaca* (strain DW4/3-1)	Peptidase, M20 family	100%
		*Cystobacter fuscus* DSM 2262	Acetylornithine deacetylase	100%
		*Stigmatella aurantiaca* (strain DW4/3-1)	Peptidase, M20/M25/M40 family	100%
Human Ro52	LEKDEREQL	*Geobacter* sp. (strain M21)	Endopeptidase La	100%
		*Seonamhaeicola marinus*	RNA polymerase sigma factor	100%
Mouse Ro52	MEMDLTMQR	*Sulfuriferula nivalis*	Phytoene synthase	88.9%
		*Corallococcus exercitus*	Phytoene/squalene synthase	88.9%
		*Corallococcus aberystwythensis*	Phytoene/squalene synthase	88.9%
		*Corallococcus* sp. CA047B	Phytoene/squalene synthase	88.9%
		*Corallococcus exercitus*	Phytoene/squalene synthase	88.9%
Mouse Ro52	KELAEKMEM	*Arenicella xantha*	RNA pol sigma factor	100%
		*Gamma proteobacterium* SS-5	RNA pol sigma factor	100%
		*Granulosicoccus antarcticus*	RNA pol sigma factor	100%
		*Gammaproteobacteria bacterium*	RNA pol sigma factor	100%
		*Granulosicoccus* sp.	RNA pol sigma factor	100%
		*Candidatus Methyloumidiphilum*	RNA pol sigma factor	100%
		*Gammaproteobacteria bacterium*	Fumarate flavoprotein	100%
		*Tindallia magadiensis*	RNA pol sigma factor	100%
		*Oceanospirillales bacterium*	RNA pol sigma factor	100%
		*Cyanobacterium* sp. IPPAS	RNA pol sigma factor	100%
		*Cyanobacterium* sp. HL-69	RNA pol sigma factor	100%
		*Culicoidibacter larvae*	RNA pol sigma factor	100%
		*Chromobacterium violaceum*	RNA pol sigma factor	100%
		*Cyanobacterium stanieri*	RNA pol sigma factor	100%
		*Clostridium cellulovorans*	RNA pol sigma factor	100%
		*Anaerolineaceae bacterium*	RNA pol sigma factor	100%
		*Pseudobythopirellula maris*	RNA pol sigma factor	100%
		*Bacteroidetes bacterium*	RNA pol sigma factor	100%
		*Epulopiscium* sp.	RNA pol sigma factor	100%
		*Betaproteobacteria bacterium*	RNA pol sigma factor	100%
		*Fulvivirga imtechensis* AK7		100%
Human M3R	AWVISFVLW	*Planctomycetes bacterium*	Uncharacterized protein	88.9%
Mouse M3R	VLWAPAILF	*Desulfobacterales bacterium*	Site-2 protease family protein	88.9%
Human La	KEALKKIIE	*Candidatus Dojkabacteria* bacterium	Uncharacterized protein	100%
		*Hydrogenimonas* sp.	Anthranilate phosphoribosyltransferase	100%
		candidate division WOR-3 bacterium	Uncharacterized protein	100%
Mouse La	QRYWQKILV	*Planctomycetes bacterium* SM23_25	Uncharacterized protein	88.9%
Mouse La	ILVDRQAKL	*Pseudomonas* sp. NFR16	Uncharacterized	100.0%
		*Pseudomonas* sp. Bc-h	Uncharacterized	100.0%
		*Pseudomonas* sp. GV021	Uncharacterized	100.0%
		*Pseudomonas abietaniphila*	Uncharacterized	100.0%
		*Pseudomonas graminis*	DUF2914 family	100.0%
		*Pseudomonas graminis*	Uncharacterized	100.0%
		*Pseudomonas graminis*	Uncharacterized	100.0%
		*Pseudomonas graminis*	DUF2914 domain	100.0%
		*Pseudomonas graminis*	Uncharacterized	100.0%
		*Pseudomonas* sp.	DUF2914 domain	100.0%
		*Pseudomonas* sp. NFACC02	Uncharacterized	100.0%
		*Pseudomonas* sp. LP_7_YM	DUF2914 domain	100.0%
		*Pseudomonas* sp. M47T1	Uncharacterized	100.0%
		*Pseudomonas eucalypticola*	DUF2914 domain	100.0%
		*Pseudomonas* sp. K1S02-6	DUF2914 domain	100.0%
Human/Mouse α-fodrin	FLSDFRDLM	*Cocleimonas flava*	Uncharacterized	88.9%
		*Verrucomicrobiales* bacterium	Uncharacterized	88.9%
		*Planctomycetaceae* bacterium	SH3 domain	88.9%

**Table 16 jcm-11-01690-t016:** Homology of predicted peptides binding to HLA alleles to viral proteins.

Allele	SjS Protein	Core Sequence	Virus	Viral Protein	Homology
HLA-DRB4*01:01	Ro52	RFLLKNLRP	Human Immunodeficiency Virus	Glycoprotein 120	100.0%
			*Serratia* phage 2050H2	Uncharacterized	87.5%
			*Klebsiella* phage 31	Endopeptidase Rz	87.5%
			*Escherichia* phage ECA2	Endopeptidase	87.5%
			*Leclercia* phage 10164RH	Uncharacterized	87.5%
			*Citrobacter* phage SH1	Endopeptidase	87.5%
			*Citrobacter* phage phiCFP-1	Uncharacterized	87.5%
			*Serratia* phage SALSA	Endopeptidase	87.5%
			*Citrobacter* phage SH2	Endopeptidase Rz	87.5%
			*Klebsiella* phage KPP-5	Endopeptidase	87.5%
			*Leclercia* phage 10164-302	Uncharacterized	87.5%
			*Enterobacter* phage E-2	Endopeptidase	87.5%
			*Klebsiella* phage NL_ZS_3	Endopeptidase Rz	87.5%
			*Serratia* phage SM9-3Y	I-spanin	87.5%
			*Escherichia* phage LL2	I-spanin	87.5%
			*Salmonella* phage phiSG-JL2	Gp18.5	87.5%
			*Yersinia* phage phiYeO3-12	Endopeptidase	87.5%
			*Enterobacter* phage E-4	Endopeptidase Rz	87.5%
			*Enterobacter* phage E-3	Endopeptidase	87.5%
			*Yersinia* phage phiYe-F10	Uncharacterized	87.5%
			*Klebsiella* phage	endopeptidase	87.5%
HLA-DRB1*01:01	Ro60	LFTFIQFKK	*Botrytis* virus X	RNA replication protein	
100%	Ro60	TYYIKEQKL	*Streptomyces* phage	Ro-like RNA binding protein	88.9%
			*Streptomyces* phage	Ro-like RNA binding protein	88.9%
			*Streptomyces* phage Beuffert	Ro-like RNA binding protein	88.9%
			*Pyramimonas orientalis* virus	Uncharacterized protein	69.2%
		KDLLRLSHL	*Botrytis* virus X	RNA replication	100%
HLA-DRB1*01:01	La	DFNVIVEAL	*Salmonella* phage SPFM12	Uncharacterized	88.9%
HLA-DRB3*01:01	La	DLDDQTCRE	*Leviviridae* sp.	RNA replicase beta chain	64.3%
HLA-DRB1*01:01	M3R	IAFLTGILA	*Bacillus* phage 031MP004	Uncharacterized	100%
			*Bacillus* phage 055SW001	Uncharacterized	100%
			*Bacillus* phage 022DV001	Uncharacterized	100%
			*Bacillus* phage 031MP002	Uncharacterized	100%
			*Bacillus* phage 031MP003	Uncharacterized	100%
HLA- DRB1*15:01	M3R	IIGNILVIV	Human immunodeficiency virus 1	Protein Vpu	100%

**Table 17 jcm-11-01690-t017:** Homology of predicted peptides binding to HLA alleles to bacterial proteins.

Allele	SjS Protein	Core Sequence	Bacteria	Bacterial Protein	Homology
HLA -DRB1*04:05	Ro52	EAGMVSFYN	*Legionella moravica*	Ankyrin	88.9%
			*Legionella* sp. Km535	Ankyrin repeat domain-containing protein	88.9%
	Ro52	MVSFYNITD	*Legionella moravica*	Ankyrin	88.9%
			*Legionella* sp. Km535	Ankyrin repeat domain-containing protein	88.9%
HLA-DRB4*01:01	Ro60	TYYIKEQKL	*Helicobacter* sp. 11S03491-1	Protoporphyrinogen oxidase	100%
			*Fusobacterium*	Uncharacterized	100%
			*Fusobacterium*	Uncharacterized	100%
HLA-DRB3*01:01	La	CREDLHILF	*Virgibacillus massiliensis*	Uncharacterized	100%
HLA-DRB4*01:01	M3R	LVTIIGNIL	Uncultured	Uncharacterized	88.9%
HLA-DRB1*01:01	Alpha Fodrin	IKLLQAQKL	*Eubacteriaceae*	Glycosyltransferase	100.0%
			*Candidatus Pseudoramibacter*	Glycosyltransferase	100.0%
HLA- DRB1*15:01	Alpha Fodrin	EVLDRWRRL	*Desulfonatronum sp.*	Thioredoxin	88.9%
			*Thermoleophilaceae bacterium*	Proline RNA ligase	88.9%
			*Thermoleophilaceae bacterium*	Proline tRNA ligase	88.9%
			*Nonomuraea nitratireducens*	DUF885 family protein	88.9%
			*Nonomuraea phyllanthi*	DUF885 domain-containing protein	88.9%
			*Firmicutes bacterium*	Biotin protein ligase	88.9%
			*Firmicutes bacterium*	Biotin protein ligase	88.9%
			*Streptomyces malaysiensis*	Putative non-ribosomal peptide synthetase	100.0%
			*Streptomyces malaysiensis*	Non-ribosomal peptide synthetase	100.0%
			*Streptomyces malaysiensis*	Carrier domain-containing protein	100.0%
			*Aquisphaera giovannonii*	Phosphomannomutase/phosphoglucomutase	100.0%
			*Streptomycetaceae bacterium*	Uncharacterized protein	88.9%
			*Curtobacterium* sp. MCPF17_047	Uncharacterized protein	100.0%
			*Nitriliruptorales bacterium*	DUF1932 domain-containing protein	100.0%
			*Paracoccus homiensis*	Acetyltransferase (GNAT) family protein	100.0%
			*Actinophytocola xanthii*	SnoaL-like domain-containing protein	100.0%
			*Frigoribacterium* sp. PhB160	S-DNA-T family DNA segregation ATPase	100.0%
			*Frigoribacterium* sp. PhB107	S-DNA-T family DNA segregation ATPase	100.0%
			*Frigoribacterium* sp. ACAM 257	Cell division protein FtsK	100.0%
			*Geodermatophilus* sp. DF01_2	Peptidase_M16_C domain-containing protein	100.0%
			*Acidobacteria bacterium*	Uncharacterized protein	100.0%
			*Nitrosococcus oceani* C-27	Transposase	88.9%
			*Nitrosococcus oceani* (strain)	Y1_Tnp domain-containing protein	88.9%
			*Dietzia* sp. MeA6-2017	Uncharacterized protein	100.0%
			*Firmicutes bacterium*	Bifunctional ligase/repressor BirA	100.0%
			*Dietzia* sp. oral taxon 368	Uncharacterized protein	100.0%
			*Saccharopolyspora* sp. ASAGF58	Uncharacterized protein	100.0%
			*Saccharopolyspora spinosa*	Uncharacterized protein	100.0%
			*Chloroflexi bacterium*	Biotin [acetyl-CoA-carboxylase] ligase	100.0%
			*Actinobacteria bacterium* 13	Biotin [acetyl-CoA-carboxylase] ligase	100.0%
			*Pelagibaca abyssi*	Uncharacterized protein	100.0%
			*Candidatus Kentron* sp. LFY	Type III restriction enzyme	88.9%
			*Planctomycetes bacterium*	Diguanylate cyclase	88.9%
			*Candidatus Kentron* sp. LFY	Type III restriction enzyme, res subunit	88.9%
			*Candidatus Solibacter* sp.	3-isopropylmalate dehydratase large subunit	88.9%
			*Hyalangium minutum*	Uncharacterized protein	88.9%
			*Actinokineospora terrae*	AraC-type DNA-binding protein	88.9%
			*Actinokineospora cianjurensis*	AraC-like DNA-binding protein	88.9%
HLA-DRB4*01:01	Alpha Fodrin	IKLLQAQKL	*Eubacteriaceae* *Candidatus Pseudoramibacter*	Glycosyltransferase Glycosyltransferase	100.0% 100.0%

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
