# Peer review of "Epitope Mapping of Pathogenic Autoantigens on Sjögren’s Syndrome-Susceptible Human Leukocyte Antigens Using In Silico Techniques"

_jcm, 2022, doi:10.3390/jcm11061690_

Round 1

Reviewer 1 Report

The Authors of this paper, by using in silico techniques and structural modelling tools, conducted a mapping of antigenic epitopes of the five most important autoantigens linked to primary Sjögren’s syndrome (pSS), i.e., Ro52, Ro60, La, M3R and α-fodrin, presented by specific pSS high-risk HLA molecules, with particular focus on HLA-DRB1*03:01, and then evaluated the molecular mimicry between such antigenic epitopes and exogenous antigens, viral and bacterial.

The paper is well written. Although it sounds very technical, which is however in the purposes and in the nature of the work itself, it is easy to approach and understandable both in the initial design and in its development and succession of sections.

The Authors well underline the limitations of the study and consequently of the results. Despite these limitations, the paper is of scientific value to the extent that, albeit using theoretical prediction models, by the complete epitope mapping of key autoantigens in pSS related to specific HLAs it sheds new light on potential molecular mimicry between such epitopes and peptides originated from exogenous agents.

The etiology of pSS is currently unknown. As hypothesized in other autoimmune diseases, it is believed that an immune dysregulation initiated and fed by possibly infectious or environmental antigens and autoantigens occurs in genetical predisposed individuals. The identification, through ad hoc experimental techniques, of antigens (purely exogenous or coming from the oral or intestinal microbiome) who cross-react with well-known pSS-related autoantigens, could be useful in clarifying also one possible aspect of the etiology of pSS. Although pSS etiology is likely multifactorial, target therapies against these agents, used at an early stage or applicable at any stage of the pSS disease if those agents result chronically involved, could represent an etiological therapy of the pSS itself. I suggest to the Authors to further emphasize this, even if hypothetical, etiological and therapeutic impact of their study, and of its further potential developments.

As a secondary aspect, please delete extra spaces present in the text.

Author Response

Rebuttal 

We would like to thank the Reviewers for your thoughtful and thorough comments on our manuscript. We have carefully considered each of them and have made corrections and updates accordingly. Below please find a detailed explanation of each change or rebuttal where necessary. The changes are tracked in the revised manuscript

Reviewer 1

The identification, through ad hoc experimental techniques, of antigens (purely exogenous or coming from the oral or intestinal microbiome) who cross-react with well-known pSS-related autoantigens, could be useful in clarifying also one possible aspect of the etiology of pSS.

We completely agreed with the reviewer that experiments need to be performed to validate or support the cross-reactive peptides or antigens identified. This pilot study determines the antigenic epitopes that are either non-cross reactive or cross-reactive with viral or bacterial antigens in-silico. We are in the process of gathering and creating reagents, mouse/human cell lines, and animal models to test this concept. The proposed ad hoc experimental techniques will take time and resources, which we honestly cannot achieve at the moment for this study. 

Although pSS etiology is likely multifactorial, target therapies against these agents, used at an early stage or applicable at any stage of the pSS disease if those agents result chronically involved, could represent an etiological therapy of the pSS itself. I suggest to the Authors to further emphasize this, even if hypothetical, etiological and therapeutic impact of their study, and of its further potential developments.

We appreciate this important suggestion. As recommended, we have expanded the discussion (lines 477-497) and added Figure 6 to support the expanded discussion.   

As a secondary aspect, please delete extra spaces present in the text.

As recommended, all extra spaces from the tables have been removed.

Reviewer 2 Report

The authors provide a huge amount of in silico data on the topic, yet there is a lack of any experimental proof. Suggest to perform the following two experiments - at least exemplified for the most abundant epitopes.

Are the identified epitopes cross-reactive with autoAbs from patients in vitro? 

One could immunize rats with the respective epitopes and use the developed antibodies to check whether they are reactive to the whole virus, bacteria, etc...in question. 

Suggest to discuss the potential geographic pattern of autoantibodies in SjS. Do certain antibodies eclusively occur in certain geographic regions as humans are exposed exclusively to certain viridae, baceteria etc...only in this region. Does the regional clustering of certain autoantibodies correlate with a specific viral or bacterial envirionment? If so I would suggest to focus on the experimental validation of these epitopes as outlined above.   (Tables 14 - 16)

Author Response

Rebuttal 

We would like to thank the Reviewers for your thoughtful and thorough comments on our manuscript. We have carefully considered each of them and have made corrections and updates accordingly. Below please find a detailed explanation of each change or rebuttal where necessary. The changes are tracked in the revised manuscript

Reviewer 2

The authors provide a huge amount of in silico data on the topic, yet there is a lack of any experimental proof. Suggest to perform the following two experiments - at least exemplified for the most abundant epitopes. Are the identified epitopes cross-reactive with auto-Abs from patients in vitro? 

We agreed with the reviewer the suggested experiment would validate the in-silico data that we presented. However, we do not have the reagents/materials. More importantly we are not certain if the suggested experiments will provide conclusive results for the following reasons: 1) we have to have access to sera of SjS patients with specific HLA alleles (HLA-DRB1*03:01, HLA-DRB1*01:01, HLA-DRB1*15:01, HLA DRB1*04:05, HLA-DRB4*01:01, and HLA-DRB3*01:01). Unfortunately, we currently do not have access to these samples and patient data, and 2) the predicted epitopes are short peptides. We and others have shown that auto-Abs from SjS patients tend to recognize conformational-dependent epitopes and possibly post-translational modification epitopes. Therefore, it is hard to predict whether auto-Abs from SjS patients will bind and react to the predicted epitopes at a quantifiably significant level. 

One could immunize rats with the respective epitopes and use the developed antibodies to check whether they are reactive to the whole virus, bacteria, etc...in question. Rats can be immunized with epitopes and their validity can be checked by observing the reactivity to the whole virus or bacteria which will require further experimental techniques.

We appreciate the reviewer’s suggestion. The predicted epitopes were identified based on the modeling of their binding interaction with specific HLAs (HLA-DRB1*03:01, HLA-DRB1*01:01, HLA-DRB1*15:01, HLA DRB1*04:05, HLA-DRB4*01:01, and HLA-DRB3*01:01). We will need humanized mice or rats with specific human HLA transgenes to perform the proposed experiment. We are discussing with Dr. Yaron Tomer (Albert Einstein College of Medicine) in acquiring the DR3-transgenic mouse strain. However, breeding and expanding this strain may take one or two years before we can perform the experiments. 

Suggest to discuss the potential geographic pattern of autoantibodies in SjS.

As suggested, we have revised Table 1 to include the autoantibodies that have been identified and associated with specific HLA in the respective patient populations. 

Do certain antibodies exclusively occur in certain geographic regions as humans are exposed exclusively to certain viridae, baceteria etc...only in this region. Does the regional clustering of certain autoantibodies correlate with a specific viral or bacterial environment? If so, I would suggest to focus on the experimental validation of these epitopes as outlined above.   (Tables 14 - 16)

The reviewer brought up two fascinating questions. As presented in the discussion, a few viruses have been shown to associate with SjS, such as HIV. This study found M3R and Ro52 epitopes to match or mimic HIV-1 proteins. We found many other viruses and bacteria however, none of them has been studied or shown to be associated with SjS.  As shown in the revised Table 1, particular HLA were identified to be associated with specific autoantibodies. However, to our knowledge, there are no published studies showing certain antibodies exclusively occur in specific geographic regions as humans are exposed solely to certain viridae, bacteria, or the regional clustering of certain autoantibodies correlate with a specific viral or bacterial environment. We strongly agreed with the reviewer that we must explore these correlations. Regarding the experimental question, as stated in the previous comment, we currently do not have the mouse/human cell lines and animal models to test this concept. 
